# Impact of Various Non-Contrast-Enhanced MRA Techniques on Lumen Visibility in Vascular Flow Models with a Surpass Evolve Flow Diverter

**DOI:** 10.3390/diagnostics14111146

**Published:** 2024-05-30

**Authors:** Yigit Ozpeynirci, Margarita Gorodezky, Augusto Fava Sanches, Sagar Mandava, Ana Beatriz Solana, Thomas Liebig

**Affiliations:** 1Institute for Diagnostic and Interventional Neuroradiology, University Hospital, Ludwig-Maximilians-University (LMU), Marchioninistr. 15, 81377 Munich, Germany; favasanches@gmail.com (A.F.S.); thomas.liebig@med.uni-muenchen.de (T.L.); 2GE HealthCare, 80807 Munich, Germany; margarita.gorodezky@ge.com (M.G.); anabea.solana@ge.com (A.B.S.); 3GE HealthCare, Atlanta, GA 30318, USA; sagar.mandava@gehealthcare.com; 4Center for Neuroimaging Sciences, King’s College London, London SE1 7EH, UK

**Keywords:** silent MRA, deep learning, flow diverter, intracranial aneurysm

## Abstract

Background: Silent MRA has shown promising results in evaluating the stents used for intracranial aneurysm treatment. A deep learning-based denoising and deranging algorithm was recently introduced by GE HealthCare. The purpose of this study was to compare the performance of several MRA techniques regarding lumen visibility in silicone models with flow diverter stents. Methods: Two Surpass Evolve stents of different sizes were implanted in two silicone tubes. The tubes were placed in separate boxes in the straight position and in two different curve configurations and connected to a pulsatile pump to construct a flow loop. Using a 3.0T MRI scanner, TOF and silent MRA images were acquired, and deep learning reconstruction was applied to the silent MRA dataset. The intraluminal signal intensity in the stent (SI_in-stent_), in the tube outside the stent (SI_vessel_), and of the background (SI_bg_) were measured for each scan. Results: The SI_in-stent_/SI_bg_ and SI_in-stent_/SI_v_ ratios were higher in the silent scans and DL-based reconstructions than in the TOF images. The stent tips created severe artefacts in the TOF images, which could not be observed in the silent scans. Conclusions: Our study demonstrates that the DL reconstruction algorithm improves the quality of the silent MRA technique in evaluating the flow diverter stent patency.

## 1. Introduction

Three-dimensional time-of-flight (TOF) magnetic resonance angiography (MRA) without contrast media is a valuable non-invasive tool for evaluating intracranial aneurysms following a flow diverter treatment [1]. However, it is limited by susceptibility artefacts that may obscure the visibility of the parent vessel lumen [1,2,3,4].

Silent MRA is a sequence that GE HealthCare has just recently introduced. This technique combines a zero echo time (ZTE) 3D radial sampling with an arterial spin-labeling preparation module [5]. The arterial spin labeling preparation is applied at the neck, tagging the flowing blood towards entering the imaging region, thus allowing to visualize the flowing blood. Prior to the labeling pulse, native images (unlabeled images) are made. After the labeling pulse, labeled images are acquired. Images of the flowing blood are produced by subtracting the labeled images from the native images in a similar fashion to digital subtraction angiography.

ZTE imaging can diminish magnetic susceptibility and minimize the phase dispersion of the labeled blood flow signal resulting in the reduction in metallic artefacts from stents or other metallic objects, such as coils, enabling the visualization of the lumen [6,7,8,9,10]. In studies investigating intracranial stents including flow diverters, the visualization ability of silent MRA was found to be superior to that of TOF MRA [6,7,8,9,10].

A deep learning-based (DL) denoising and deranging algorithm was recently introduced to the market by GE HealthCare as AIR^TM^ Recon DL (GE HealthCare, Waukesha, WI), which includes a deep convolutional neural network to support the reconstruction process of raw data, thus increasing the signal-to-noise ratio and producing clean and sharp images [11]. The purpose of this study was to compare the performance of several non-contrast-enhanced MRA approaches regarding lumen visibility in flow models using the Surpass Evolve flow diverter.

This is the first in vitro study evaluating the abilities of deep learning-based silent MRA sequence in the imaging of a flow diverter stent.

## 2. Materials and Methods

### 2.1. Flow Diverter

Surpass Evolve (SE, Stryker Neurovascular, Kalamazoo, MI, USA) is a flow diverter stent used in the endovascular embolization of intracranial aneurysms. It consists of 64 braided cobalt chromium wires interwoven with platinum–tungsten wires to improve visibility under fluoroscopy.

### 2.2. Experimental Setup

Under fluoroscopic guidance, two stents of different sizes (width × length) (4.5 × 25 mm^2^ and 5 × 20 mm^2^, respectively) were implanted in two silicone tubes with an inner diameter of 4.4 mm (Figure 1). To confirm the optimal wall apposition of the stents, a contrast-enhanced flat panel CT was performed.

Three boxes were created for each scan set. The silicone tubes (one without a stent and two with each stent) were placed in three separate transparent plastic boxes in the straight position for the first series of scans and then in two different curve configurations (obtuse angle curve (C1)) and (almost a) right angle curve (C2)) for the second and third series of scans, respectively.

The water-filled boxes and silicone tubes were connected to a pulsatile flow pump to construct a flow loop. The Flowcon1000 v1.5 software (αCandis GmbH, Pforzheim, Germany) was utilized to control the pump.

The T1 relaxation time of the circulating water was reduced by adding gadolinium contrast material, 0.7 mmol of Magnevist (Bayer Vital GmbH, Leverkusen, Germany), in the closed water circuit of about 7 L to simulate the T1 characteristics of the blood (T1 ≈ 1.6 s). A Coriolis flowmeter (PROMASS 83F08, Endress+Hauser GmbH, Reinach, Switzerland), used to measure a ground-truth net flow value with a precision of ±0.5 mL/s, was connected after the outlet of the models close to the pump.

The constructed object was then positioned in the center of the body coil with the longitudinal axis of the tubes parallel to the main magnetic field. The acquisitions were conducted under pulsatile flow conditions (60 beats/min). For the confirmation of pulsatility of the flow and to trigger the sequences, a pulse oximeter was attached to the silicone tube before entering the box. The mean flow rate was set to 400 mL/min, as determined by the flowmeter, simulating the flow conditions of the internal carotid artery.

The setup can be observed in Figure 2 and Figure 3.

### 2.3. MR Imaging

MR images were obtained with a GE HealthCare 3.0T MRI scanner (Discovery MR750w, GE Healthcare, Milwaukee, WI, USA), using the coils integrated in the scanner table. At first, we obtained 3D TOF MRA with parameters as follows: spoiled gradient-echo sequence with TR/TE/FA, 24 ms/3.4 ms/15°; FOV/acq matrix, 200 mm × 200 mm/200 × 200; acq slice thickness, 1 mm, reconstructed voxel size, 0.5 × 0.78 × 0.78 mm^3^; and ARC = 2. A vein saturation slab was not used. The number of slices was 150. The scan time was 2 min and 40 s.

The parameters for the silent MRA were as follows: ASL-prep 3D radial ZTE sequence with TR/TE/FA, 3.077 ms/0.028 ms/5°; FOV/acq matrix, 180 mm × 180 mm/180 × 180; and reconstructed voxel size, 1 × 0.7 × 0.7 mm^3^. The number of slices was 120. The scan time was 4 min and 12 s.

### 2.4. Image Post-Processing and Analysis

A labeled database of 10,000 image pairs, representing near-perfect and conventional MRI images, was used to train a DL model in a supervised manner [11]. The conventional training data were synthesized from near-perfect images using established methods to create lower resolution versions with more truncation artifacts and with higher noise levels. The model was designed to remove the noise seen in ZTE scans and improve the perception of image resolution. This trained model was used to reconstruct silent MRA raw data, with the acquired data being processed through both the conventional approach and the DL-based approach.

ImageJ Version 1.54 (US National Institutes of Health, Bethesda, MD, USA) was used for the image analysis. The signal intensities inside the tubes and of the background were measured. The measurements were conducted in the plane perpendicular to the course of the tubes at three locations along the segments with the stents (both stent ends and in the middle) and two areas outside of the stent (5 mm proximally and distally to the stent ends) using circular regions of interest (ROIs). The mean value of the signal intensity measurements from the stent-free areas are represented by the vessel SI (SI_v_) and that from the stent segments are represented by the in-stent SI (SI_in-stent_).

The signal intensity of the background (SI_bg_) was measured in the plane through the midsection of the stents. The ratios of SI_in-stent_ to SI_bg_ and of SI_in-stent_ to SI_v_ were calculated.

## 3. Results

The SI_in-stent_/SI_bg_ and SI_in-stent_/SI_v_ ratios were higher in both the silent MRA scans with and without DL-based reconstructions than in the TOF images in models with the straight and C1 configurations. However, this trend could not be consistently seen on images in models with C2 configuration. SI_in-stent_/SI_bg_ showed the strength of the DL denoising algorithm in minimizing the background noise in all configurations and with and without a stent (Table 1 and Table 2).

The stent tips created severe artefacts on the TOF, which could not be observed in the silent scans (Figure 4, Figure 5 and Figure 6).

## 4. Discussion

Our research sought to assess the efficacy of different non-contrast-enhanced MR angiography methods in the evaluation of the stent lumen in vascular flow models carrying the Surpass Evolve flow diverter. In the models with varied curves and two different stent sizes, intraluminal SIs were measured in the areas with and without a stent.

The SI_in-stent_/SI_bg_ and SI_in-stent_/SI_v_ ratios were consistently higher in the silent scans and DL-based reconstructions than in the TOF, except in the models with a tighter curve. The stent tips caused severe artefacts in the TOF compared to the silent MRA.

One of the advantages of the silent MRA is the reduction in metallic artefacts from stents or other metallic objects, such as coils or surgical clips due to the ZTE imaging, enabling a better visualization of blood flow [5,6,7,8,9,10]. Additionally, the negative effect of the turbulent flow observed in TOF MRA may diminish [5,6,7,8,9,10]. In studies evaluating intracranial laser-cut and braided stents, the visualization capability of the silent MRA was found to be superior to that of 3D TOF MRA. In addition, the silent MRA was able to reveal neck remnants of coiled aneurysms with a greater accuracy than the TOF. The silent MRA could thus identify aneurysm occlusion more effectively than the TOF. Moreover, the silent MRA demonstrated a superior visualization of aneurysms treated with a flow diverter stent regardless of the aneurysm location, the degree of in-stent stenosis, and the presence or absence of additional coiling [5,6,7,8,9,10].

Further advantages of silent MRA are insensitivity to motion artefacts due to radial sampling and the elimination of noise, increasing patient comfort and resulting in less anxiety in the scanner with better compliance [12,13].

However, as Holdsworth et al. [5] also stated, silent MRA has some intrinsic problems, mainly due to the known limitations of the arterial spin labeling technique in the regions of slow flow. In their study, they observed poorly defined and smaller caliber vessels suggesting wall irregularities in the silent scans, most probably related to the slower flow along the vessel wall compared to the center. A second issue they pointed out was the attenuation or absence of patent vessels. In another study, in 5 of 27 silent intracranial MRAs, distal vessels could not be evaluated due to a poor inflow [14]. Therefore, caution was advised, since it could mimic vascular pathologies.

We observed this finding especially in the models with a tighter curve (C2) (Figure 6). In the model without a stent, the intraluminal signal was overall very poor but poorer in the regions where the tube enters and exits the box. In the models with stents, there was almost no signal in the stent. One explanation for this could be the slow flow caused by a kinking or stenoses of the silicone tubes at the entry or exit points of the boxes. Another explanation for the poorer results in the C2 models could be that the labelling location, which was placed 10 cm below the imaging volume, tagged less flowing water in the model with respect to the other models, decreasing the signal within the tube. A potential reason for the poorer results could also be the model’s instability within the phantom, causing movement with each pulse. This movement likely affected the silent MRA, as it relies on reference measurements obtained by subtracting two images.

The DL-based denoising algorithm could partially overcome the limitations of the silent MRA scan in these regions. In most of the models, the SI_in-stent_ was higher after DL reconstruction compared to the silent MRA scans. DL could enhance the stent signal and, at some points, restore the signal loss, which could especially be appreciated in models with a tighter curve (Figure 6). The silent MRA scans in this study use a self-calibrated coil sensitivity method that can lead to sub-optimal coil sensitivity estimates, especially in regions with very low signal intensities. The use of a dedicated external calibration is likely to improve performance and will be explored in future work.

The last thing to discuss is the absence of artefacts at the stent tips in the silent MRA compared to the severe artefacts seen in the TOF sequences in the same regions.

A more pronounced signal loss at the ends of stents is typically observed with stents that contain additional radiopaque markers at their tips, which raises the quantity of metal and increases susceptibility and radiofrequency artefacts [4,15]. However, Surpass Evolve does not have extra radiopaque markers at its tips. Moreover, this phenomenon was not related to the configuration of the curve, as it has been demonstrated that artefacts may increase as the angle between the magnetic field and stent orientation grows [16].

The turbulent flow pattern seen at the edges of the stent may explain the focal signal loss on the TOF. Additionally, exposed loop wires at the stent tips can exacerbate eddy currents, resulting in increased radiofrequency shielding the artefacts [17].

The reported better background signal suppression and visualization of blood flow in regions with a turbulent pattern [18] with the silent MR imaging technique could explain the absence of this artefact.

Our limitations were as follows:

Each of the models were scanned independently and, although we tried to build the models similarly and place them similarly in the MRI scanner, the experimental conditions were not exactly equivalent.

Our models’ magnetic field interactions are unable to represent the interactions between intracranial vessels and their surroundings. As a result of the experiment’s geometry, variations in the labeling’s magnitude relative to a human are to be anticipated. To surmount the limitations of the present investigation, in vivo research must be conducted.

We evaluated models that lacked aneurysms. Not investigated was the interaction between the aneurysm and the flow diverter stent. Therefore, this investigation cannot draw a conclusion regarding the status of aneurysm occlusion evaluation.

## 5. Conclusions

Our study demonstrated that DL-based denoising reconstruction may overcome the limitations of the silent MRA technique and be useful in the evaluation of the stent patency after a flow diverter treatment. To investigate the significance of this sequence in the follow-up of aneurysms treated with stents, additional in vitro and in vivo research is required.

## Figures and Tables

**Figure 1 diagnostics-14-01146-f001:**
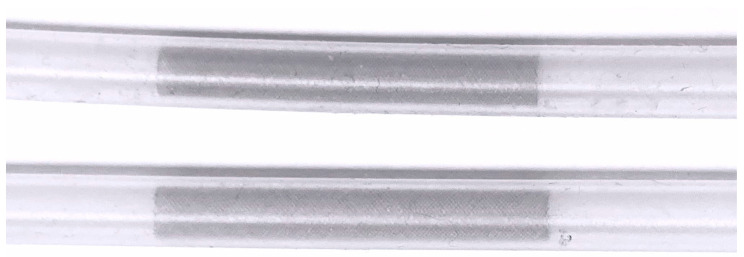
A close-up of the silicone tubes.

**Figure 2 diagnostics-14-01146-f002:**
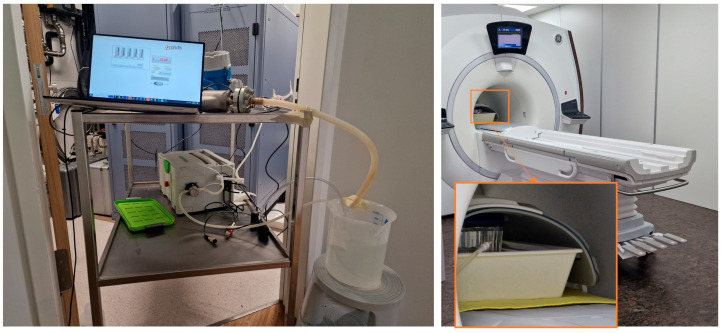
Flow phantom setup demonstrating the scanner (**right**) and the pump and the flowmeter in the control area (**left**).

**Figure 3 diagnostics-14-01146-f003:**
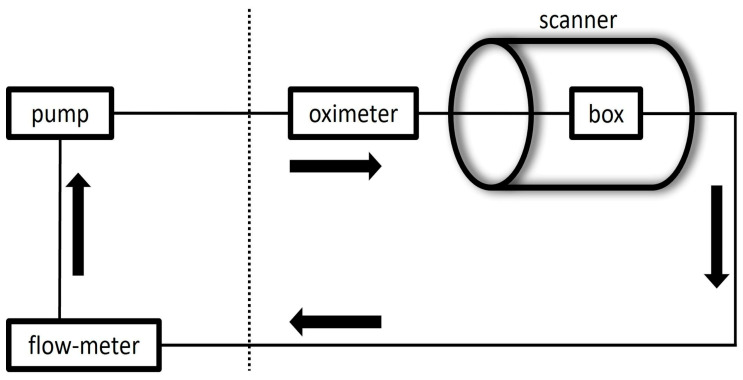
Schematic representation of the experimental setup. Dashed line represents the separation between the control and magnet areas.

**Figure 4 diagnostics-14-01146-f004:**
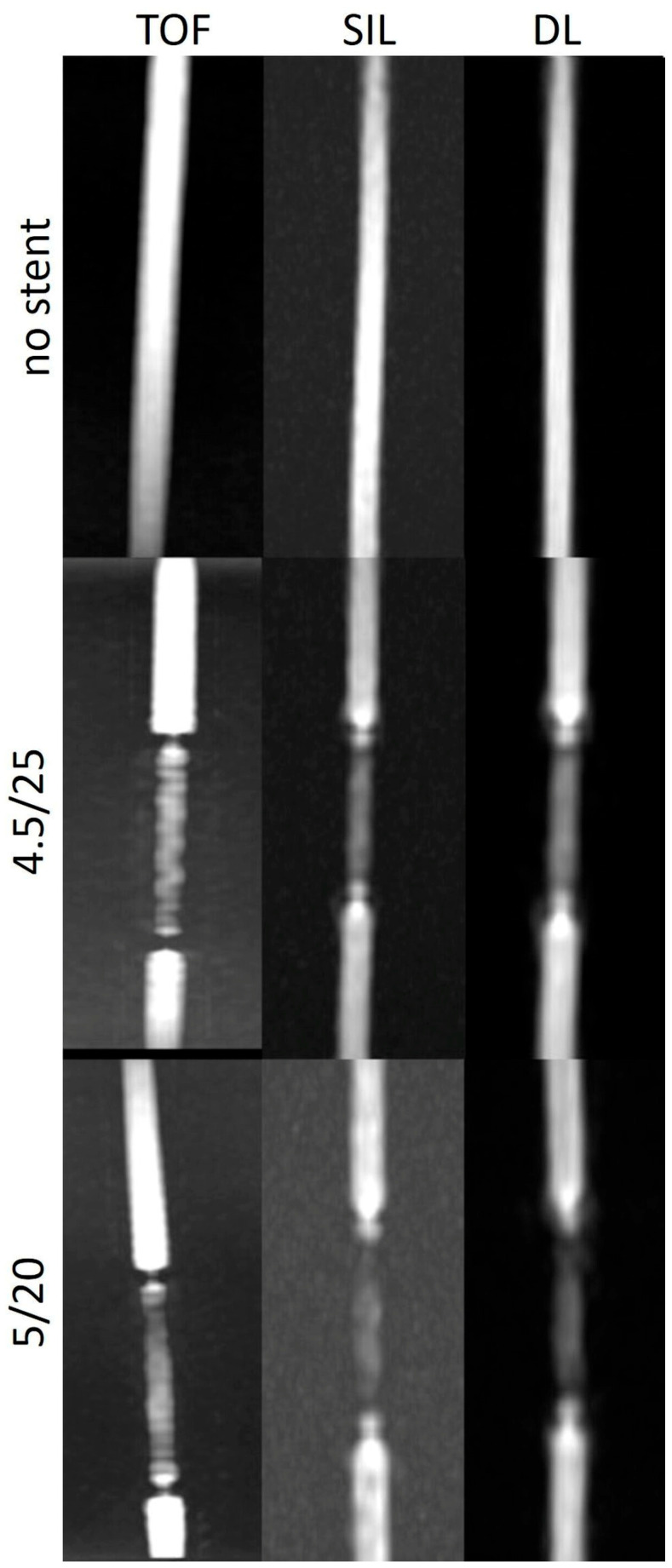
Three-dimensional maximum intensity projection images of straight models with and without stents acquired using different non-contrast-enhanced MRA techniques. TOF: time of flight; SIL: silent sequence; DL: deep learning.

**Figure 5 diagnostics-14-01146-f005:**
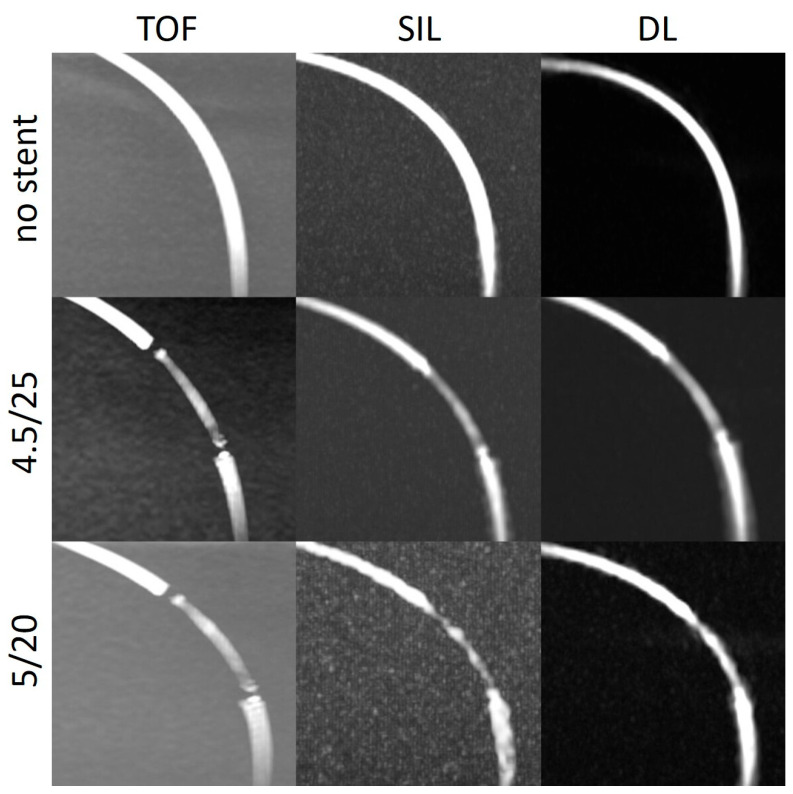
Three-dimensional maximum intensity projection images of the curved models (obtuse angle, C1) with and without stents acquired using different non-contrast-enhanced MRA techniques. TOF: time of flight; SIL: silent sequence; DL: deep learning.

**Figure 6 diagnostics-14-01146-f006:**
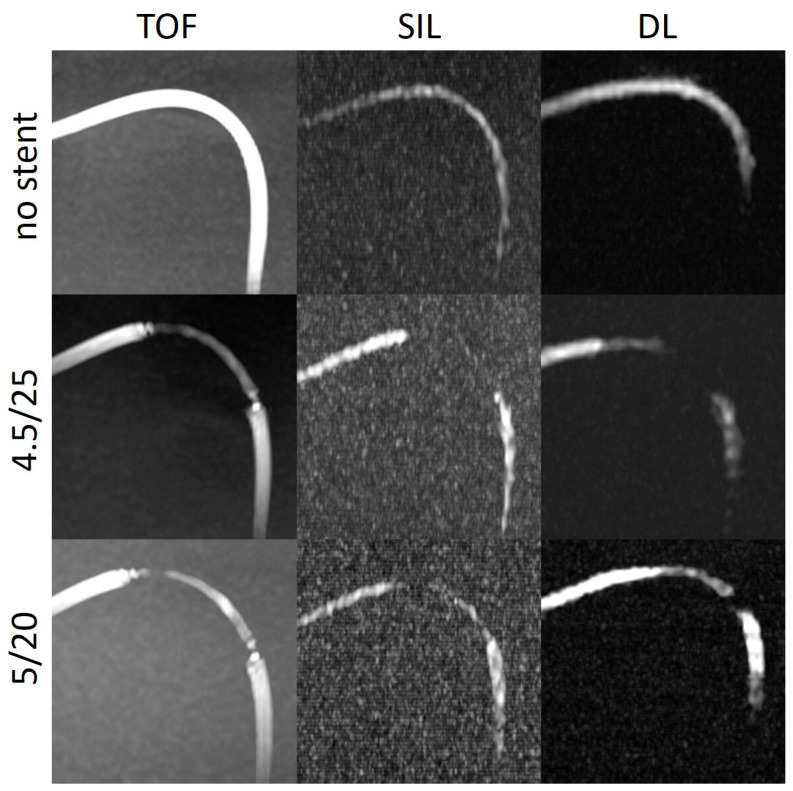
Three-dimensional maximum intensity projection images of the curved models (right angle, C2) with and without stents acquired using different non-contrast-enhanced MRA techniques. TOF: time of flight; SIL: silent sequence; DL: deep learning.

**Table 1 diagnostics-14-01146-t001:** Ratio of the mean signal intensity in the stent (SI_in-stent_) to the signal intensity of the background (SI_bg_).

SI_in-stent_/SI_bg_			
	TOF	SIL	DL
STR			
No	5.3	7.1	80.1
4.5/25	2.0	6.1	57.1
5/20	2.4	3.3	35.3
C1			
No	7.1	7.1	72.2
4.5/25	1.8	5.8	75.3
5/20	2.3	2.2	22.3
C2			
No	8.7	2.1	20.2
4.5/25	5.1	1.5	10.4
5/20	2.2	1.4	8.0

TOF: time of flight; SIL: silent sequence; DL: deep learning; STR: straight; C1: curve1; C2: curve2; no: without stent.

**Table 2 diagnostics-14-01146-t002:** Ratio of the mean signal intensity in the stent (SI_in-stent_) to the mean signal intensity of the vessel (SI_v_).

SI_in-stent_/SI_v_			
	TOF	SIL	DL
STR			
No	1.0	1.0	1.0
4.5/25	0.5	1.0	0.9
5/20	0.4	0.9	0.9
C1			
No	1.1	1.1	1.1
4.5/25	0.4	0.9	1.0
5/20	0.4	0.8	0.8
C2			
No	1.5	1.0	1.2
4.5/25	0.7	0.8	0.6
5/20	0.8	0.9	0.5

TOF: time of flight; SIL: silent sequence; DL: deep learning; STR: straight; C1: curve1; C2: curve2; no: without stent.

## Data Availability

The data presented in this study are available upon reasonable request from the corresponding author.

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
