# Peer review of "Impact of Various Non-Contrast-Enhanced MRA Techniques on Lumen Visibility in Vascular Flow Models with a Surpass Evolve Flow Diverter"

_diagnostics, 2024, doi:10.3390/diagnostics14111146_

Round 1

Reviewer 1 Report

Comments and Suggestions for Authors

Can the use of head coil improve the results? Can the use of contrast enhanced sequences improve  the results? Please discuss this.

There are stll limitations in SIL/ DL results as shown  shown in the pictures at the end of the stents. 

Author Response

We appreciate your precious time in reviewing our paper and providing valuable comments. 

Can the use of head coil improve the results? Can the use of contrast enhanced sequences improve the results? Please discuss this. 

Author response: Thank you for pointing this out. While we do not expect a significant impact on the use of a head coil versus a torso coil in general for the phantom setup, there would be of course differences between different coils related to channel geometry, density, coverage, penetration etc. A coil that covers exactly the field of view that wants to be imaged and it is closer to the phantom would provide the best results, above all, as the silent MRA is a radial acquisition and will benefit from reduced signal outside of the field of view of interest as streaking artifacts will be diminished. 

While the use of contrast dye could potentially enhance the results, the authors aimed to focus this paper on noncontrast angiography sequences.  

There are still limitations in SIL/ DL results as shown in the pictures at the end of the stents. 

Author response: We acknowledge that this limitation is present in TOF images. There may indeed be partial volume effects and dephasing due to blood acceleration at the ends of the stents. However, when examining the simplest models (straight), it is evident that the most significant effects are not at the ends of the stents in DL or SIL images. 

Reviewer 2 Report

Comments and Suggestions for Authors

The study evaluated the performance of 2 non-contrast enhanced MRA techniques, including silent MRA and TOF MRA, specifically focusing on their impact on lumen visibility in vascular flow models using the Surpass Evolve flow diverter. The study found that deep learning could improve the quality of the silent MRA in evaluating the flow diverter stent patency.

The results in the manuscript mention, "The SIin-stent/SIbg and SIin-stent/SIv ratio were higher on silent scans and DL-based reconstructions than on TOF images." However, TOF images visually appeared superior to DL in Figure 6  (C2 model) , especially vessel in the 4.5/25 DL image was interrupted and mostly unclear. Yet, according to Table 1, the score of DL (C2 model) was higher than that of TOF. This result can be misleading. The current method of calculating the signal within the stent places too much emphasis on the signals at both ends of the stent, failing to reflect the overall visibility of the stent segment. Therefore, the evaluation methods used in the study need improvement.

In clinical applications, most flow diverters might involve the paraclinoid segment of the internal carotid artery, where the carotid might make a 180-degree turn. This is likely the most common application in clinical settings. However, this scenario was not studied in the article.

Author Response

We appreciate you taking the time to review our paper and provide valuable comments. 

The results in the manuscript mention, "The SIin-stent/SIbg and SIin-stent/SIv ratio were higher on silent scans and DL-based reconstructions than on TOF images." However, TOF images visually appeared superior to DL in Figure 6 (C2 model), especially vessel in the 4.5/25 DL image was interrupted and mostly unclear. Yet, according to Table 1, the score of DL (C2 model) was higher than that of TOF. This result can be misleading. The current method of calculating the signal within the stent places too much emphasis on the signals at both ends of the stent, failing to reflect the overall visibility of the stent segment. Therefore, the evaluation methods used in the study need improvement. 

Author response: Figures 4, 5, and 6 present 3D maximum intensity projection (MIP) images. However, measurements were conducted on the source images. This discrepancy may explain the differences between the visual appearance of the models on MIP images and the actual signal-to-noise intensity measurements and ratios.   

The following is added to the Discussion section: A potential reason for the poorer results could also be the model's instability within the phantom, causing movement with each pulse. This movement likely affected the silent MRA, as it relies on reference measurements obtained by subtracting two images. 

In clinical applications, most flow diverters might involve the paraclinoid segment of the internal carotid artery, where the carotid might make a 180-degree turn. This is likely the most common application in clinical settings. However, this scenario was not studied in the article. 

Author response: Thank you for highlighting this point. It would indeed be interesting to explore the capabilities of DL-based sequences in models with a 180-degree turn, as well as in models representing pathologies such as aneurysms or stenosis. As we continue to develop our models, we anticipate publishing more data. 

While we have not yet employed a 180-degree model, our study suggests that the results, though not statistically proven, tend to be independent of model geometry. Therefore, we might expect that such a model would yield similar results, which could be confirmed in future research alongside more complex aneurysm geometries. 

Reviewer 3 Report

Comments and Suggestions for Authors

The paper is presented well and is of good quality.

A few specific comments are noted below

Reference

Description

Fig 1

Maybe replace this with an image of the three tubes in the phantom if possible as described in line 71-74

Fig 2

The photo appears to be of low resolution. Can this be improved as it’s almost impossible to see what is happening

Can the authors comment on the possibility/practicality of combining TOF and silent sequences to

Line 112

Can the authors explain a little more about the training so don’t have to go to the reference? Just one or two lines.

Line 122

“The mean value of measurements” did you mean “The mean value of signal intensity measurements”

Author Response

We greatly appreciate your valuable time in reviewing our paper and offering insightful comments. 

Maybe replace Fig1 with an image of the three tubes in the phantom if possible as described in line 71-74. 

Author response: Unfortunately, we did not capture a picture of the phantoms.  

Fig2 appears to be of low resolution. Can this be improved as it’s almost impossible to see what is happening? 

Author response: Thank you for bringing this to our attention. The Fig2 is replaced with a new, higher-resolution version.   

Can the authors comment on the possibility/practicality of combining TOF and silent sequences too? 

Author response: We apologize for any confusion. If the question pertains to combining two sequences at the scanner, we don't believe this is feasible due to their distinct technical backgrounds. Furthermore, we couldn't find any publications discussing such a combination. 

However, if the question refers to performing two sequences sequentially for a specific clinical purpose, we acknowledge the possibility but note that it would significantly extend scan time. In such cases, we would prefer to conduct one of these sequences and, if necessary, consider adding contrast-enhanced MRA for clarification in uncertain situations.  

Can the authors explain a little more about the training so don’t have to go to the reference? Just one or two lines (line 112). 

Author response: The revised text in the "2.4. Image Post-Processing and Analysis" section, with changes highlighted in bold, reads as follows:  

A labeled database of 10000 image pairs, representing near-perfect and conventional MRI images, was used to train a DL model in a supervised manner [11]. The conventional training data were synthesized from near-perfect images using established methods to create lower resolution versions with more truncation artifacts and with higher noise levels. The model is designed to remove noise seen in ZTE scans and improve the perception of image resolution. This trained model was used to reconstruct silent MRA raw data, with the acquired data being processed through both the conventional approach and the DL based approach. 

“The mean value of measurements” did you mean “The mean value of signal intensity measurements” (line 122)?  

Author response: The text has been updated accordingly.